# Sorting polymerization in a bichannel metal-organic framework

Keat Beamsley [ID], Nobuhiko Hosono [ID] ✉ & Takashi Uemura [ID] ✉

Accomplishing multiple synthetic tasks in parallel, including substrate capture, separation, and reaction, along with controlled arrangement of product, all in one system has remained a long-standing challenge in synthetic chemistry. Here, we report a sorting polymerization strategy that harnesses the multifunctional nature of a bichannel metal-organic framework (MOF). The MOF, [Cu(5-methylisophthalate)]$_n$, featuring two distinct one-dimensional channels arranged in a single Kagome lattice, allows selective adsorption of monomers to different sites based on their polarity and size. This enables the sorting of different vinyl monomers and their in-situ parallel homopolymerization within the respective channels. The process produces alternating single-chain arrays of homopolymers in a single step, a configuration unattainable by conventional approaches. Additionally, the introduction of inter-chain cross-linking allows for the isolation of the binary polymer array by removing the MOF template. This work highlights the potential of MOFs as versatile reaction platforms for the synthesis of complex, well-ordered molecular architectures from chaotic mixtures of raw materials.

Developing a reaction system programmed to selectively recognize and bind target molecules, drive specific reactions, and assemble the resulting products into well-defined structures has been a major challenge in synthetic chemistry. Metal–organic frameworks (MOFs), composed of metal ions and organic linkers assembled into crystalline porous networks, have attracted considerable attention due to their tunable pore structures and versatile chemical functionalities[1–4]. Over the past few decades, MOFs have been extensively studied as promising adsorbents for gas storage and separation[5–9], owing to their highly porous nature and the ability to adapt their frameworks to interact with targeted guest molecules[10–14]. More recently, interest has expanded toward exploiting MOFs not only as passive adsorbents but also as reaction environments, in which their well-defined nanospaces can act as templates exhibiting precise control over guest reactions[15–20]. Despite these advancements, the two primary functions of MOFs—selective molecular uptake and control over the reactions of guests—have predominantly been developed independently of one another. By integrating both of these features into a single system, we anticipated that the multiple operations of guest recognition, compartmentalization, and reaction control previously unattainable in conventional synthetic systems could be realized.

Here, we present a synthesis method, termed 'sorting polymerization,' which integrates the dual functionalities of MOFs to achieve specific molecular capture, compartmentalize guests, and perform mutually isolated reactions on specific components from a mixed feed of substrate monomers within a single system. Conventional MOF-promoted reactions typically employ frameworks with a single type of pore, which limits their ability to recognize, arrange, and promote reactions of multiple different compounds in parallel[21–25]. In this study, we utilized a 'bichannel' MOF featuring two distinct one-dimensional (1D) channels arranged in a Kagome lattice (Fig. 1a)[26]. The MOF, [Cu(mip)]$_n$ (mip = 5-methylisophthalate), referred to hereafter as **1**, consists of an alternating pattern of large hydrophobic channels and smaller hydrophilic channels separated by impermeable channel walls. These channels provide selective environments that accommodate specific molecules based on their polarity and size, enabling the simultaneous sorting of two different vinyl monomers and their homopolymerization within designated channels in a single step (Fig. 1b).

Department of Applied Chemistry, Graduate School of Engineering, The University of Tokyo, 7-3-1 Hongo, Bunkyo-ku 113-8656 Tokyo, Japan.
✉ e-mail: nhosono@g.ecc.u-tokyo.ac.jp; uemurat@g.ecc.u-tokyo.ac.jp

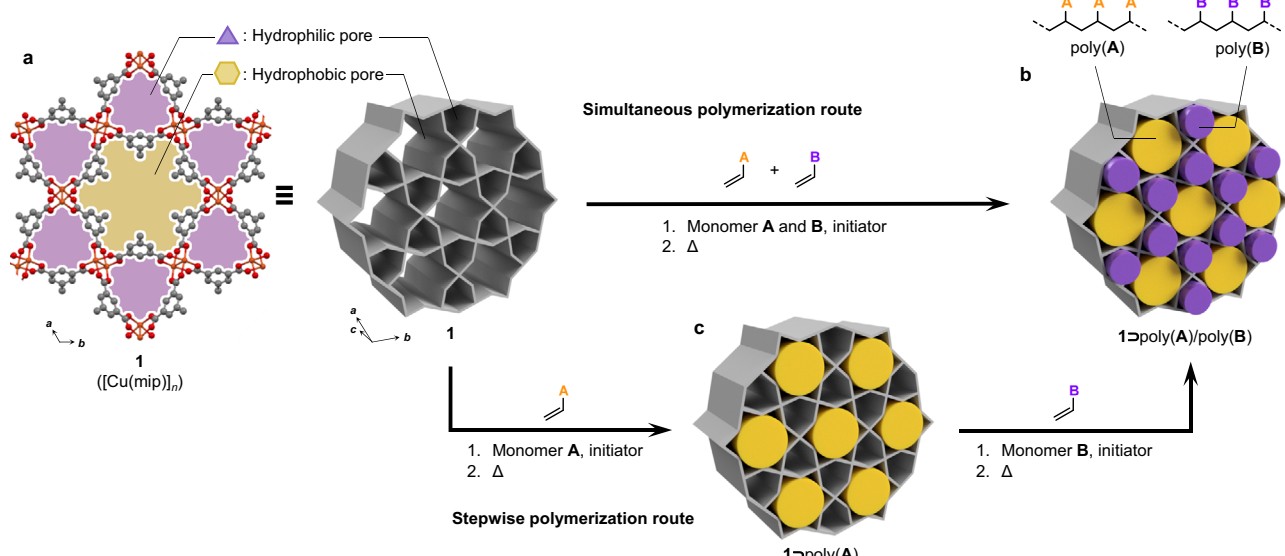

**Fig. 1 | Sorting polymerization in a bichannel MOF: simultaneous route and stepwise route. a** Crystal structure of **1**, which possesses two different types of channels: a small hydrophilic channel (purple triangular area) and large hydrophobic channel (yellow hexagonal area). **b** Simultaneous route. Simultaneous incorporation of large hydrophobic monomer **A** and small hydrophilic monomer **B** together with a radical initiator, followed by in situ polymerization by heating, provides an array structure of separate single-chains of poly(**A**) and poly(**B**) in **1**, namely **1**⊃poly(**A**)/poly(**B**). **c** Stepwise route. Incorporation of monomer **A** and the radical initiator in the hexagonal channel, followed by in situ polymerization by heating, provides **1**⊃poly(**A**). Successive incorporation of small hydrophilic monomer **B** and the initiator in the triangular channel, followed by in situ polymerization by heating, provides **1**⊃poly(**A**)/poly(**B**).

Alternatively, the monomer incorporation and polymerization can be conducted stepwise, allowing confirmation of the compartmentalized formation of different polymers within the respective MOF channels (Fig. 1c). Furthermore, this monomer sorting approach enables the formation of 1D alternating single-chain arrays of two distinct polymers—a structural configuration that is unattainable through conventional synthetic techniques. This sorting strategy not only exemplifies the synergy between molecular recognition and controlled reactions in confined spaces but also provides a new approach to producing well-defined macromolecules directly from a mixture of different monomers or even reactive contaminants.

Furthermore, by incorporating cross-linking ligands into **1** via a mixed-ligand strategy, the alternating single-chain polymer array structure was permanently stabilized, enabling its isolation as discrete particles following the removal of the MOF template. The interchain alternating arrangement produced in this study represents a distinct configuration in which two homopolymers are alternately aligned at the molecular level. By integrating the molecular sorting capabilities of MOFs with compartmentalized polymerization techniques, this approach lays a foundation for the creation of highly engineered polymeric materials with promising applications in molecular electronics, energy conversion, and beyond[27–30].

## Results

### Channel-selective adsorption in 1
The bichannel structure of **1** is comprised of independent triangular and hexagonal 1D channels, with diameters ($d$) of 0.4 nm and 0.9 nm, respectively, oriented along the $c$-axis. These channels are completely separated by the framework's structural walls, which consist of mip ligands interconnected by Cu(II) paddle-wheel clusters at the corners (Fig. 1a)[26,31]. The triangular channel features more polar and hydrophilic pore surfaces compared to the hexagonal channel, due to axial $H_2O$ coordination sites on the paddle-wheel clusters, which point exclusively into these triangular pores[32,33]. This observation led to the hypothesis that selective monomer incorporation based on polarity could be achieved. Indeed, an isoreticular analog of this

MOF, STAM-1, has been reported to exhibit selective gas adsorption in its two distinct pores, depending on the polarity of the adsorbates[33]. Furthermore, single-crystal X-ray diffraction analysis of solvent-loaded STAM-1 has demonstrated that solvent molecules bearing coordinative moieties preferentially interact with the axial coordination sites of the paddle-wheel clusters, highlighting the potential for pore-selective inclusion of specific guests through coordination-driven interactions[34].

To confirm channel-selective adsorption in **1**, we synthesized the framework via conventional solvothermal methods using $mipH_2$ and copper(II) acetate (see Methods, Supplementary Fig. 1) and performed $N_2$ and $H_2O$ vapor sorption studies. The $N_2$ adsorption isotherm at 77 K exhibited a typical Type-I profile (Fig. 2a), while the $H_2O$ adsorption isotherm at 298 K displayed a gate-type profile, indicative of a structural transition from a closed to an open phase, consistent with analogous frameworks (Fig. 2b). Grand Canonical Monte−Carlo (GCMC) simulations of adsorption isotherms where sorption was constrained to the hexagonal channel for $N_2$ and the triangular channel for $H_2O$ were carried out, which successfully provided isotherm profiles consistent with the experimental data and reflected the observations for the related MOF, STAM-1 (Supplementary Fig. 2, Supplementary Methods 2 and 3)[32,33]. The results agreed best with the adsorption of $N_2$ exclusively in the hexagonal pore, and $H_2O$ in the triangular pore of **1**, supporting the presence of polarity-dependent channel-selective adsorption in **1** (see Methods).

### Monomer sorting and polymerization in 1
The polarity-based channel-selective adsorption observed in **1** was extended to include vinyl monomer guests, which were polymerized at the positions they were sorted into within the MOF. Several vinyl monomers of varying molecular sizes and polarities were used: styrene (**S**), methyl vinyl ketone (**MVK**), acrolein (**Ac**), acrylamide (**AAm**), and *N*-vinylformamide (**NVF**). These monomers were selected to cover a broad range of polarities, thereby enhancing the desired sorting effects. Initially, each monomer type was investigated separately in individual experiments.

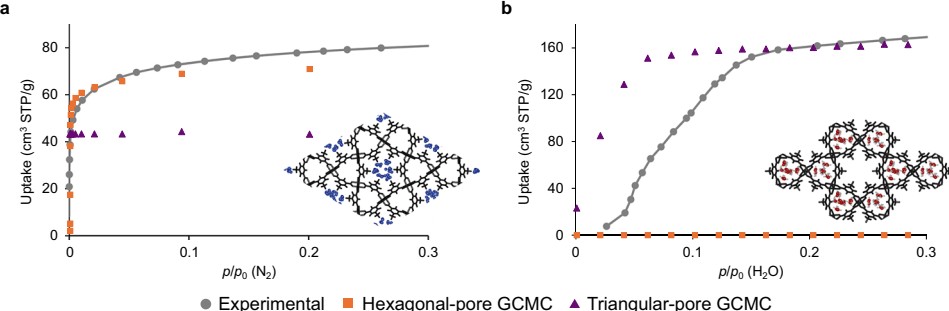

**Fig. 2 | Channel-selective adsorption in 1. a**, **b** the experimental adsorption isotherms of N₂ at 77 K (**a**) and H₂O at 298 K (**b**). Experimental isotherm (grey circle with solid line), simulated isotherm by GCMC with adsorption constrained to hexagonal channels (orange square), and simulated isotherm by GCMC with adsorption constrained to triangular channels (purple triangle). The inset figures show simulation snapshots for the adsorption states best matching experiments: N₂ in the hexagonal channel (**a**) and H₂O in the triangular channel (**b**). The deviation of the simulated H₂O isotherm from the experimental data by a shift in the inflexion point was due to a gate-opening structural transition upon adsorption, which was not modeled in the presented fixed-phase simulation. Likewise, deviations at higher $p/p_0$ values for both adsorbates are attributed to external surface sorption not modeled in GCMC.

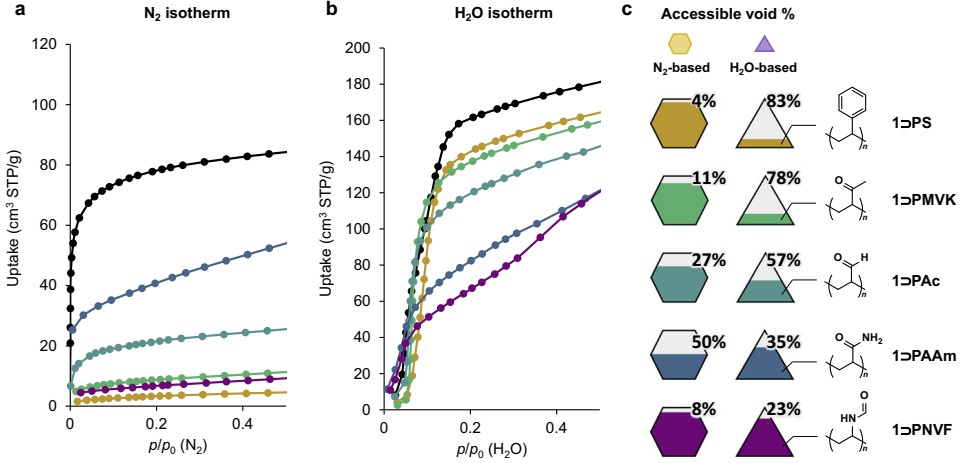

**Fig. 3 | N₂ and H₂O adsorption isotherms of the composites. a** N₂ adsorption isotherm at 77 K (**a**) and H₂O adsorption isotherm at 298 K (**b**) for the **1**-polymer composites. Empty **1** (black), **1⊃PS** (khaki), **1⊃PMVK** (green), **1⊃PAc** (teal), **1⊃PAAm** (blue), and **1⊃PNVF** (purple). **c** Accessible void percentages of the respective hexagonal and triangular pores, calculated using the relative decrease of the micropore adsorption amount in the N₂ (**a**) and H₂O (**b**) isotherms, respectively.

The monomer and a radical initiator were introduced either by bulk soaking of **1** crystals in the liquid monomer followed by evaporation of excess liquid, or, for non-volatile monomers, by slow evaporation of the solvent from a monomer solution, leaving the monomer within the framework (see Methods and Supplementary Method 1). The resulting composite, **1⊃M** (M: monomer), was heated at a minimum of 70 °C for at least 48 hours to induce polymerization, forming a **1**-polymer composite, **1⊃PM** (PM: poly(M)) (see Supplementary Table 1 for system-specific initiator and temperature conditions). This process was repeated with monomers of varying polarities to produce corresponding MOF-polymer composites, designated as **1⊃PS, 1⊃PMVK, 1⊃PAc, 1⊃PAAm**, and **1⊃PNVF**. Each composite was characterized by scanning electron microscopy (SEM) and powder X-ray diffraction (PXRD), which confirmed the retention of the original crystal morphology and framework structure post-polymerization (Supplementary Figs. 3 and 4). ¹H NMR analysis of decomposed aliquots from both pre- and post-polymerization composites was employed to determine the monomer conversion and resultant polymer loading amount in MOF in each experiment (Supplementary Fig. 5). Overall, high monomer conversion was achieved for all monomers, ranging from 46.3 to 99.5%, corresponding to the polymer loading amount of 56-133 mg/g (Supplementary Table 1). The

molecular weights (MWs) of the polymers synthesized within **1** were determined using size-exclusion chromatography (SEC) on the digested samples, which ranged from 1000 to 3000 g mol⁻¹ (Supplementary Table 2).

Built upon the above results, it was anticipated that **S**, with an approximate size of 0.7 nm, would be accommodated exclusively within the hexagonal channel ($d$ = 0.9 nm) due to its size incompatibility with the smaller triangular channel ($d$ = 0.4 nm), as well as its non-polar nature. In contrast, the smaller and relatively polar monomers, **MVK, Ac, AAm**, and **NVF**, were expected to be introduced into both hexagonal and triangular channels, resulting in the occupation of both channels by the respective polymers following the polymerization reaction.

The selective polymer occupation within one of the two pore types was determined through adsorption measurements for the composite, using N₂ and H₂O as probes for the hexagonal pore volume and triangular pore volume, respectively (Fig. 3). The uptake of each probe in the micropore region ($p/p_0$ = 0–0.2) decreased relative to that of the empty **1** (Fig. 3a, b), with the percentage decrease attributed to the filling of the corresponding pore by the polymer. The adsorption isotherms were analyzed to determine the total adsorption volume for each probe molecule using the Brunauer–Emmett–Teller (BET)

equation for N$_2$, and the co-operative sorption equation of Dalby et al. for H$_2$O (Supplementary Fig. 6)[35]. Using **1⊃PS** as an example, the adsorption volume of N$_2$ in the empty **1** was 66 cm$^3$ (STP)/g, whereas after the polymerization of **S** within the pores, the adsorption volume decreased significantly to 2.9 cm$^3$ (STP)/g (Fig. 3a). This substantial reduction indicates that **PS** was formed within the hexagonal pores. In contrast, the adsorption volume of H$_2$O vapor remained largely unchanged before and after polymerization (Fig. 3b), suggesting that **PS** was not introduced into the triangular pores. Hence, the quantity of each of the two residual void types in **1** can be estimated based on the reduction in the adsorption volume of the respective probe molecules. Although complete blockage of gas diffusion pathways and an increase in composite mass by the polymers may lead to an underestimation of the void volumes, this approach nonetheless provides an independent measure of polymer occupation within each pore type (Fig. 3c).

The structural flexibility of **1** also confirms the incorporation of polymers within the pores. The original empty **1** exhibits a reversible temperature-induced structural transition from an 'open' to 'closed' phase upon heating to over 100 °C, accompanied by a loss of mass to desorption of H$_2$O (Supplementary Figs. 7 and 8). However, once polymers are incorporated into the pores, **1** can become resistant to temperature-induced changes. A similar effect of polymeric guests on the flexibility of MOFs has been observed in other flexible MOF systems[36], reinforcing the conclusion that polymers have formed within the nanopores of **1**. Interestingly, the phase transition behavior in the current system exhibits polymer dependence (Supplementary Figs. 9–11). Upon heating, **1⊃PS** undergoes the phase transition around 100 °C (Supplementary Fig. 9), whereas **1⊃PNVF** permanently retains the open-phase structure over 150 °C (Supplementary Fig. 10). This difference can be attributed to the presence of **PNVF** in the triangular channels, where Cu(II) coordination sites on the paddle-wheel clusters are exposed to the pore surfaces. **PNVF** coordinates to these open metal sites, preventing the transformation required for the paddle-wheel clusters to adopt the closed phase[31,37] and thus keeping the framework open even at higher temperatures. In contrast, **PS**, which occupies only the hexagonal channels, does not induce this effect.

It was observed that as the hydrophilicity of the monomer decreases, the polymer product tends to shift from occupying both pores to preferentially filling the hexagonal (hydrophobic) pore over the triangular (hydrophilic) pore (Fig. 3a,b). In extreme cases, such as with **S** and **MVK**, polymerization within the pores renders the MOF completely incapable of N$_2$ sorption while maintaining its H$_2$O sorption capacity. This phenomenon is attributed to the concept of an 'immobilized adsorbate'—wherein polymerized monomers become fixed in locations with high affinity, blocking those sites and leaving low-affinity locations available for adsorbates of opposite polarity. Consequently, **S** and **MVK** are the most suitable monomers for inducing selective polymerization exclusively within the hexagonal channel, leaving the triangular pore unoccupied after the initial polymerization (Fig. 3c). The vacant triangular pore can then be utilized for a subsequent polymerization with smaller, polar monomers, such as **AAm** and **NVF**.

## Alternating single-chain array in 1 via simultaneous and stepwise polymerization

Having confirmed the existence of monomer species which are polymerized in one or both pores respectively, the possibility arises to realise simultaneous monomer sorting followed by parallel homopolymerization within the two channels. To validate this concept, a mixture of **S** and **NVF** was introduced and polymerized simultaneously in **1** using 2,2′-azobis(isobutyronitrile) (AIBN) as radical initiator (Fig. 1, see Methods). The amount of **S** and **NVF** monomers introduced was 10.7 wt% and 7.1 wt%, respectively (Supplementary Table 3). The latter was adjusted to be significantly less than the previously determined capacity for **NVF** alone (13.4 wt%) (see Supplementary Method 1 and

Supplementary Table 1). Given this composition, we anticipated that all **NVF** monomers would be directed into the triangular channels through strong coordination-based adsorption, thereby preventing co-adsorption with **S** in the larger hexagonal channels. The polymerization reaction was initiated by gradually increasing the temperature from 70 °C to 150 °C. This simultaneous polymerization approach achieved high conversion of 89.4% and 100% for **S** and **NVF**, respectively (Supplementary Table 3), successfully filling both pores with loadings of 9.6 wt% and 7.1 wt% for **PS** and **PNVF** to give the composite **1⊃PS/PNVF**. Further, acid digestion of the composite allowed for isolation of the two respective homopolymers **PS** and **PNVF**, corroborating the segregated polymerization of the monomers within **1** (Supplementary Figs. 12–14). Notably, no copolymers were detected, despite **S** and **NVF** being a copolymerizable pair with reactivity ratios of 10.1 and 0.34, respectively (Supplementary Fig. 15)[38]. Thus, **1** selectively directs **S** and **NVF** monomers into the hexagonal and triangular channels, respectively, where they subsequently undergo in situ polymerization to afford their respective homopolymers, even when starting from a mixture of monomers. However, in this approach, the location of each polymer type could not be discerned easily as sorption may only determine which pores are filled, not which polymers are filling them (Supplementary Fig. 16). Therefore, the polymerization was split into a stepwise method where the hexagonal channels would first be filled with **PS**, and then the remaining triangular channels filled with **PNVF**, allowing determination of the location of the two polymers by studying the differences between composites before and after introduction of each in two steps (Fig. 1, see Methods).

In the first step, **S** and AIBN were incorporated into the hexagonal channels of **1** by soaking the crystals in a solution of AIBN dissolved directly in monomer, followed by the removal of excess monomer under reduced pressure, yielding **1⊃S**. The composite was heated by gradually increasing the temperature from 70 °C to 150 °C to initiate polymerization. $^1$H NMR analysis of the resulting composite, **1⊃PS**, indicated a monomer conversion of 71.3% and a **PS** content of 11.8 wt% (Supplementary Fig. 12, Supplementary Table 4). N$_2$ and H$_2$O adsorption measurements on **1⊃PS** revealed that **PS** was exclusively formed within the hexagonal pores, leaving the triangular pores vacant (Supplementary Fig. 16). Subsequently, a second polymerization was conducted by reintroducing **NVF** and AIBN into the vacant triangular pores of **1⊃PS** and applying the same temperature program to yield **1⊃PS/PNVF**. $^1$H NMR analysis of the process showed **PNVF** formation in the composite with the **PNVF** content of 7.6 wt%. N$_2$ and H$_2$O adsorption tests of the final product confirmed that both the hexagonal and triangular pores were filled (Supplementary Fig. 16). Isosteric heats of H$_2$O adsorption further supported site-selectivity of polymers for their respective pores (Supplementary Figs. 17 and 18), and variable-temperature FT-IR spectra point to coordination of PNVF's amide groups to Cu in the triangular pores as a driving force for this selectivity (Supplementary Fig. 19). While the peak positions in PXRD patterns remained largely unchanged throughout the process, changes in peak intensity were observed due to the incorporation of guest molecules into the pores, indicating that the MOF's crystal structure was maintained but that the distribution of electron density within the crystal system may have changed upon formation of polymer inside (Supplementary Fig. 13).

To verify the stepwise selective polymerization, electron density maps of the composite were generated using the Maximum Entropy Method (MEM) based on PXRD data at each stage (Fig. 4, see Methods, Supplementary Figs. 20–22, Supplementary Method 4)[39,40]. Although atomic resolution of the framework was not achieved because of technical limitations related to the inherent disorder of the guest polymers and distribution of ligand orientation of **1**, the electron density of both the framework itself and that of polymeric guests located inside was able to be visualized. Overlap of symmetry-related

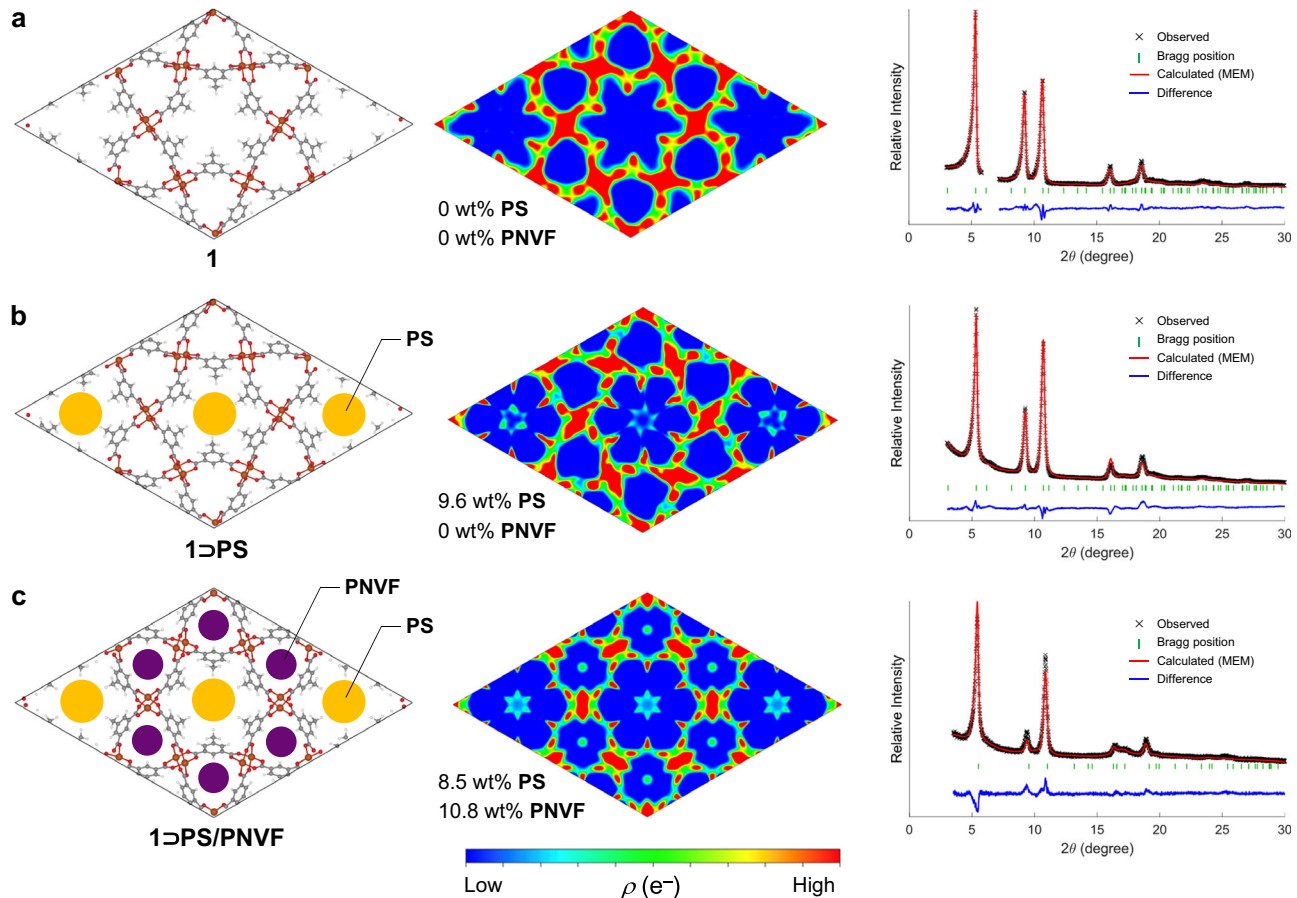

**Fig. 4 | Stepwise polymerization of different monomers in binary channels. a–c** Schematic illustrations of the filling pattern (left), electron distribution maps (center), and results of MEM analysis (right) for empty **1** (**a**), **1⊃PS** (**b**), and **1⊃PS/PNVF** (**c**) (Supplementary Figs. 20–22).

peaks in powder data resulted in guest density shapes more strongly influenced by crystal symmetry than by meaningful orientation information, but the quality was nevertheless sufficient to confirm which of the two pores–hexagonal or triangular - contained polymers within the MOF. In the pre-polymerization sample, no electron density corresponding to guest molecules was observed in either the hexagonal or triangular pores (Fig. 4a). However, after the polymerization of **S**, a significant increase in electron density was detected in the hexagonal pores, indicating the formation of **PS** within these pores (Fig. 4b). After the second polymerization step with **NVF**, electron density was observed in the triangular pores, confirming the formation of **PNVF** in these channels (Fig. 4c). These results demonstrate the successful stepwise introduction of different polymers into distinct pores. Given that the pore sizes of **1** can accommodate only a single chain of the respective polymers, each polymer is expected to adopt 1D extended form within the respective pores. These findings suggest that an alternating single-chain polymer array can be generated using the bichannel structure of **1** as a template.

**Cross-linking and isolation of alternating polymer array**
Considering the intriguing alternating array structure of single polymer chains, we further investigated the possibility of isolating this structure by removing the parent MOF. However, isolating such an array proved challenging, as the polymers tend to separate and revert to their randomly oriented bulk states upon extraction or dissolution of **1**. To address this limitation, we extended the system to incorporate cross-linking between the alternating polymer chains, thereby stabilizing the array structure. This was accomplished by incorporating the cross-linking ligand, 2,5-divinylisophthalic acid, (dvipH$_2$) into the

framework synthesis, replacing a portion of the original linkers, mipH$_2$. This produced [Cu(dvip)$_c$(mip)$_{1-c}$]$_n$, hereafter referred to as **1$_{xc}$**, where $c$ denotes the actual incorporation rate of the divinyl linker (dvip) in the MOF. The mixed-ligand **1$_{xc}$** was synthesized under the same conditions as **1**, with varying dvipH$_2$ compositions, from 3 to 100 mol% (see Methods, Supplementary Table 5). All compositions up to at least 60 mol% yielded crystalline products with PXRD patterns identical to that of the parent **1**, indicating the successful synthesis of **1$_{xc}$** (Supplementary Fig. 23). The incorporation rates of dvip were confirmed by $^1$H NMR analysis of the digested materials, verifying the incorporation of the cross-linking ligand at the ratio of 1.4 to 50.5 mol% ($c = 0.01–0.51$), respectively, thereby resulting in **1$_{x0.01}$**, **1$_{x0.04}$**, **1$_{x0.05}$**, **1$_{x0.10}$**, **1$_{x0.27}$**, and **1$_{x0.51}$** (Supplementary Fig. 24, Supplementary Table 5). The optimized host template, **1$_{x0.10}$**, was selected for the subsequent polymerization reactions for its balance of high dvip content and high pore capacity.

Similar to **1**, the respective pores of **1$_{x0.10}$** were filled with **PS** and **PNVF** in a stepwise manner. However, in this case, the polymers incorporated vinyl groups from the 5- and 2- positions of dvip, respectively (Fig. 5). This resulted in crosslinks through the channel walls, forming a frustrated crosslinked network of polymer chains with opposite polarity that were prevented from phase separation, even upon removal of the MOF template. N$_2$ and H$_2$O adsorption studies confirmed the sequential filling of the two distinct pores, and $^1$H NMR analysis indicated conversion rates of 77.0% for **S** and 100% for **NVF**, as well as 62.0% for dvip embedded in the MOF walls (Supplementary Table 5). The polymer loadings in the final composite were determined to be 8.5 wt% for **PS** and 10.8 wt% for **PNVF**. PXRD and SEM analyses confirmed the structural integrity of the host **1$_{x0.10}$** throughout the

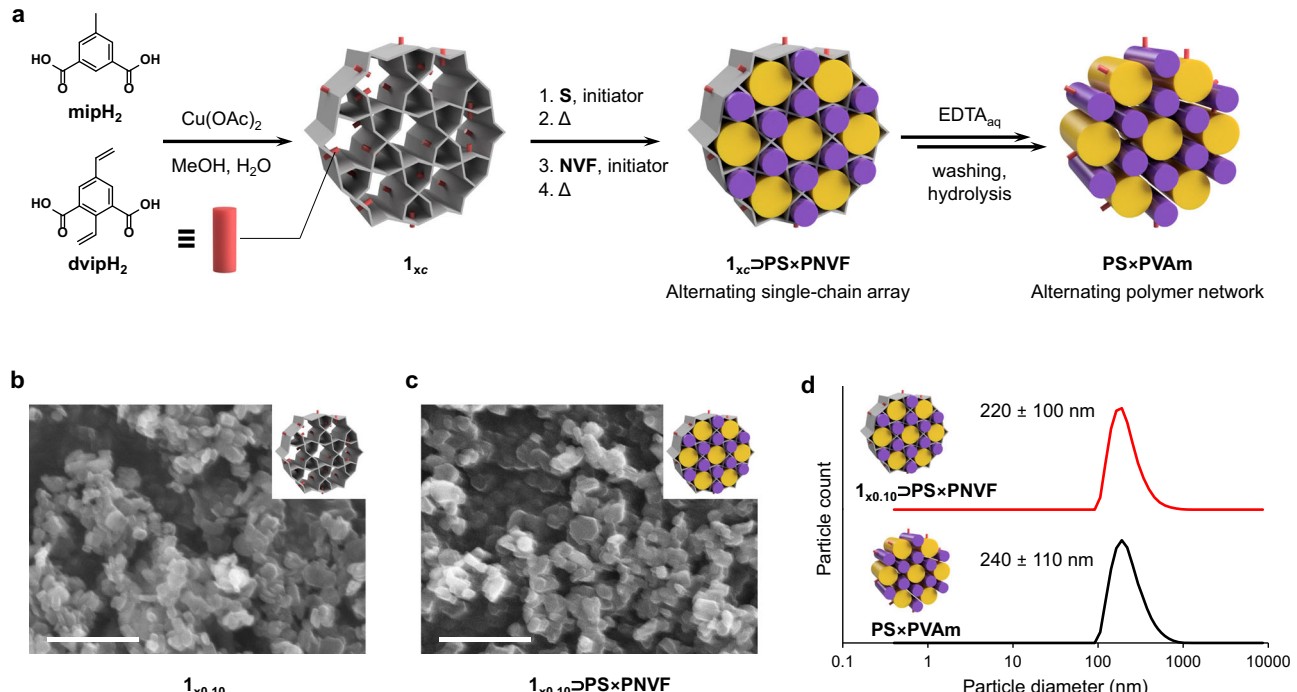

**Fig. 5 | Synthesis of alternating polymer network. a** Synthetic scheme of **PS×PVAm** alternating polymer network via a cross-linkable MOF template, **1ₓ**. **b, c** SEM micrographs of the pristine **1ₓ0.10** (**b**) and **1ₓ0.10⊃PS×PNVF** obtained after the stepwise polymerization of **S** and **NVF** (**c**). Scale bar: 1 μm. **d** number-average particle size distribution for **1ₓ0.10⊃PS×PNVF** (red) and **PS×PVAm** (black) obtained by MOF template dissolution and base hydrolysis of polymer, measured by DLS in water at 25 °C. The particle sizes were presented as mean ± standard deviation.

cross-linking polymerization process, strongly indicating the formation of a crosslinked alternating polymer network, namely **1ₓ0.10⊃PS×PNVF** (Fig. 5). The cross-linked polymer array formed in the MOF was then isolated using aqueous ethylenediaminetetraacetic acid (EDTA) solution to digest the framework, and post-synthetic modification was also demonstrated by hydrolysis of **PNVF** segments to polyvinylamine (**PVAm**).

Importantly, even after digesting the **1** template using aqueous EDTA solution, the presence of insoluble particles matching the size of the original **1** crystals was observed through dynamic light scattering (DLS) measurements (Fig. 5d). This suggests the formation of an alternating single-chain array stabilized by interchain cross-linking via dvip ligands, which persists even upon template removal. Thermal analysis of the isolated material after collection by centrifugation showed no glass transition for either **PS, PVAm**, or **PNVF** homopolymers, even after multiple heating and cooling cycles, indicating that the cross-linking had stabilized the polymer array against phase separation (Supplementary Fig. 25). Solid-state NMR measurements verified the presence of both **PS** and **PVAm**, with relaxation times suggesting both intimate mixing of the two on the nanoscale and a decrease in chain mobility presumed to be due to the presence of crosslinks (Supplementary Figs. 26 and 27)[41–43]. PXRD measurements initially showed no significant diffraction, likely due to the frustrated and non-crystalline nature of the **PS** and **PVAm** chains, which create difficulties in the detection of any superlattice formed by the alternating polymer chains. However, addition of HCl to the material caused a broad peak near $2\theta = \sim 4°$ to appear (Supplementary Fig. 28). This likely indicates the emergence of the ordered array structure, assisted by electrostatic repulsion between polymer chains following the protonation of **PVAm**, as well as enhanced electron density contrast due to the intercalation of chloride ions. This finding underscores a responsive characteristic in the structural organization of **PS×PVAm**, potentially representing a unique property derived from the incorporation of the polyelectrolyte **PVAm** as one of the two array components.

## Discussion

We developed a monomer sorting polymerization strategy using a bichannel MOF that has two distinct pore environments. By tailoring the pore dimensions and surface functionalities, we achieved selective uptake of two different monomers into separate channels, followed by in situ polymerization to form their respective homopolymers from a mixed feed. An alternative, stepwise polymerization approach further confirmed that each respective channel plays host to a separate, compartmentalized reaction. Additionally, the sorting polymerization in this bichannel MOF yielded an alternating single-chain polymer array that is not accessible through conventional polymerization methods. Incorporating a cross-linking ligand into the MOF structure then enabled the formation of a crosslinked alternating polymer network that remained intact even after the MOF template was removed, resulting in an insoluble alternating array of distinct polymer chains. These findings demonstrate how MOFs can be engineered to integrate multiple synthetic tasks, including molecular recognition, separation, reaction, and assembly of the resulting products, within a single system. This approach offers a blueprint for directly producing complex polymer materials from mixed feeds. Further, the ability to control polymer arrangement at the single-chain level represents a breakthrough in the construction of nanostructured devices, combining the softness of polymers with the sub-nanoscale regularity typically reserved for crystalline systems, which is especially relevant for applications in molecular electronics and energy conversion technologies. Extending the concept to other monomers, polymerization mechanisms, and MOF architectures holds great potential for next-generation material synthesis, enabling multiple, molecular-level precision operations within a single platform.

## Methods

### General instruments

Solution-state $^1H$ nuclear magnetic resonance (NMR) spectra were recorded using a Bruker Avance III HD spectrometer equipped with a PABBO probe operating at 500 MHz. Solid-state $^{13}C$-$\{^1H\}$ NMR spectra were recorded using a double-resonance 4 mm magic-angle spinning probe with a 9.4 T Bruker Avance III 400 MHz spectrometer. All PXRD data were recorded on a Rigaku model SmartLab X-ray diffractometer using Cu Kα radiation in Bragg-Brentano (BB) reflection mode unless otherwise stated. VT-PXRD utilized an XRD-DSC stage attachment with samples under $N_2$ flow. PXRD data for MEM analysis utilized a capillary rotation module in parallel-beam Debye-Scherrer mode. The scan rate was 0.256 deg/min, data sampling 20 points/min and rotation rate 100 rpm. Analytical SEC measurements were performed using two polystyrene gel columns in series (Shodex KF-806M and KD-802) at 40 °C on a SHIMADZU model LC-2050 system equipped with refractive index (RI) and ultraviolet (UV) detectors. The mobile phase was LiBr/dimethylformamide (DMF) 10 mM solution at a flow rate of 1.0 mL/min. SEM measurements were performed using a Hitachi model SU-5000 at an accelerating voltage of 15 kV. Samples were deposited on a conducting carbon tape attached to a SEM sample holder, then coated with osmium. $N_2$ gas adsorption and $H_2O$ vapor adsorption measurements were performed volumetrically using a MicrotracBEL model BELSORP-mini and BELSORP-aqua3, respectively. The samples were dried and evacuated at 150 °C for 16 h under high vacuum (<10 Pa) prior to the measurements. DLS measurements were performed using a Malvern model Zetasizer equipped with a He-Ne laser ($\lambda = 633$ nm) at an angle of 173° at 25 °C. Thermogravimetric analysis was carried out on a Rigaku Thermo plus EVO2 TG-DTA8122 thermogravimetric analyzer under continuous nitrogen flow. Differential scanning calorimetry (DSC) was carried out using Hitachi High-Tech Science Corporation model DSC7020 at the heating rate of 10 K/min. Variable-temperature diffuse-reflectance infrared Fourier-transform spectroscopy (DRIFTS) was carried out under continuous nitrogen flow on a JASCO FT/IR-4200 equipped with a DR-600Ci variable-temperature DRIFTS attachment. Electrospray ionization time-of-flight mass spectrometry was performed on a Bruker Q-TOF mass spectrometer with MeOH as eluent.

### Materials

All reagents and chemicals used in this study were obtained from, FUJIFILM Wako Pure Chemicals, and Tokyo Chemical Industry, unless otherwise noted. Deuterated solvents for NMR spectroscopy were purchased from Kanto Chemical. AIBN and APS were recrystallized from methanol and ethanol solutions, respectively. **S, Ac, MVK**, and **NVF** were purified by vacuum distillation prior to use. **AAm** was purified by recrystallization from acetone prior to use.

### GCMC simulations

The framework structures of **1** in closed and open phases were modeled based on the reported crystal structure of STAM-17-OEt[26,31]. STAM-17-OEt is one of several isoreticular MOFs with the same Kagome topology as STAM-1 and **1**. We used STAM-17-OEt as the parent model framework since its single-crystal X-ray structural data in both its open and closed phases have been previously reported[31]. The pre-optimization open and closed-phase structures of **1** based on these data were subjected to geometry optimization using DFT calculations with cell parameters from Le Bail analysis (Supplementary Fig. 2, Supplementary Methods 2 and 3). The DFT-optimized model frameworks, which matched experimental PXRD patterns well, were then used in successive GCMC simulations and were also used to provide phase information for PXRD reflections in MEM analysis. Electrostatic charges for use in GCMC simulations were determined using the REPEAT method[44] as implemented in CP2K (Supplementary Method 3)[45].

GCMC simulations were carried out in the RASPA2 software package[46] using fixed-geometry DFT-optimized closed- and open-phase models as described earlier. Electrostatic interactions were evaluated using Ewald summation. Lennard-Jones interactions were modeled using parameters from UFF for Cu, and DREIDING-UT[47] for C, O, and H. Lennard-Jones interactions were calculated with a spherical cutoff of 12 Å, and tail-corrections[48]. Interactions between different types of atoms were calculated by the Lorentz-Berthelot mixing rule. As rigid gate-open and gate-closed models were concluded by Sławek et al.[33] to be sufficient to model sorption in the related MOF, STAM-1, flexible force fields were not used in this study. The closed-phase simulations were run on a $1 \times 1 \times 5$ unit-cell supercell, while open-phase simulations were run on an overall $2 \times 2 \times 4$ supercell consisting of a $2 \times 2 \times 1$ 'replica cell' array of $1 \times 1 \times 4$ independently modelled super-cells—this was to save computational resources while still ensuring all supercell dimensions are over 24 Å to prevent self-overlap of atomic van der Waals cutoff radii, and to prevent limited-space effects. Each simulation run consisted of at least 100,000 initialization cycles and 200,000 simulation cycles, with each cycle consisting of $N$ Monte–Carlo moves, where $N$ is either the number of molecules in the system, or 20, whichever is greater. Multiple runs were performed at different pressures and fixed temperature, giving each point on a simulated adsorption isotherm. The purpose of the initialization steps is to give the system time to approach equilibrium from an initial state where the MOF is empty, before recording adsorbate loading information. The adsorption loading—both the final average value given for adsorption at a given temperature and pressure, and the error based on the statistical distribution of data samples—were recorded in the latter simulation step.

Adsorbate $H_2O$ molecules were modeled by the popular Tip5p-Ew model[49], while $N_2$ modeled by Martín-Calvo et al. was used as provided in the RASPA2 software package[46]. Tip5p-Ew is a rigid five-point model for $H_2O$ consisting of a chargeless van der Waals (vdW) interacting center O, two vdW-inert H atoms bearing positive partial charges, and two vdW-inert dummy atoms bearing negative charges to represent lone pairs (the total charge adds to 0). The $N_2$ model consists of vdW-interacting negatively charged N atoms rigidly bound to a central vdW-inert dummy atom bearing a positive charge such that the overall charge is neutral. In these ways, partial charges were considered in the adsorbate models to reproduce H-bonding and coordination effects in water, and quadrupole effects in nitrogen.

### Synthesis of 1

**1** was synthesized by using the literature procedure[26]. 3.6 g of mipH$_2$ (20 mmol) and 4.0 g of copper(II) acetate monohydrate (20 mmol) were mixed in 40 mL of MeOH and 20 mL of $H_2O$. After sonication of the mixture for 10 min, at which point a light blue precipitate had already begun to form, it was heated in a Teflon-lined stainless autoclave at 110 °C for 4 days. Upon cooling to room temperature, the suspension was filtered to recover a light blue solid. This solid was then sonicated for 10 min in MeOH, and filtered once more. The solid was then sonicated in acetone for 10 min again, filtered, and washed with MeOH/acetone to remove any residues from the pores. The light blue solid was finally thoroughly activated by vacuum drying at 150 °C for 16 h prior to use. The PXRD pattern of **1** was in good agreement with the simulated pattern based on an isoreticular MOF, STAM-17-OEt, which was used as a basis to develop full structural models for **1** (Supplementary Figs. 1 and 2, Supplementary Method 3)[31]. The Brunauer–Emmett–Teller (BET) surface area of **1**, calculated from the $N_2$ adsorption isotherm at 77 K, was determined to be 288.5 m$^2$/g. This value is somewhat larger than the reported BET surface area for the related framework STAM-1, supporting the larger size of hexagonal channels in **1**, which allow for incorporation of large guests such as **S**[50].

## Synthesis of $1_{xc}$

The divinyl linker (dvipH$_2$) was synthesized as described in Supplementary Method 5. dvipH$_2$ was incorporated into the standard **1** synthesis in the feed ratios of 3.0, 6.0, 9.0, 18.0, 30.0, 60.0, and 100 mol% by replacing a corresponding amount of mip in the synthetic solution of **1**. As it was found that overnight sonication of the synthetic solution gave crystallites with negligibly different crystallinity and morphology from the solvothermal method, these milder conditions were employed in this case to mitigate any undesired autopolymerization of the vinyl linkers in the solution. The light blue precipitates obtained were washed vigorously with methanol and acetone, then vacuum dried at 150 °C for 16 h. The crystallinity of the samples was checked by PXRD, and the actual dvip incorporation rates were determined by $^1$H NMR to be 1.4, 3.6, 5.0, 9.7, 26.9, 50.5, and 100.0 mol% ($c$ = 0.01-1.00), respectively (Supplementary Figs. 23 and 24). PXRD patterns matching that of pristine **1** confirm the formation of $1_{x0.01}$, $1_{x0.04}$, $1_{x0.05}$, $1_{x0.10}$, $1_{x0.27}$, and $1_{x0.51}$ with matching topology, but $c$ = 1.00 gave a different non-porous structure (referred to as $2_{x1.00}$) (Supplementary Fig. 23). Of these, N$_2$ and H$_2$O adsorption isotherms only up to $c$ = 0.10 were comparable to those of **1**, indicating no pores were blocked at these concentrations of dvip in the framework (Supplementary Table 5).

## Single-monomer and two-monomer simultaneous insertion polymerization in 1

The insertion methods were tailored for each monomer based on their volatility and physical state. For the volatile liquid monomers **S**, **MVK**, and **Ac**, the initiator was dissolved directly in the monomer, which was then introduced into **1** by soaking for 30 min under sonication. Subsequently, the bulk phase of the monomer was removed by evaporation at a pressure not lower than 0.7 times the monomer's vapor pressure. In contrast, for the non-volatile monomers **AAm** (solid) and **NVF** (high-boiling liquid), acetone solutions containing the initiator were prepared, soaked into **1** for 30 min under sonication, and then evaporated. Because a single cycle of this process often left excess monomer on the external surface of **1** crystals, acetone was reintroduced and evaporated repeatedly (2–3 cycles) until no residual monomer was observed.

For volatile monomers, the loading amount was regulated by controlling the evaporation pressure to remain below the threshold for bulk condensation but above the threshold for desorption from the micropores, thus selectively removing only monomer outside the pores. In contrast, the loading of non-volatile monomers was determined by the ratio of monomer mass to the mass of **1**, generally set to achieve approximately 0.1 mL of monomer per gram of **1**, assuming the monomer density is equal to its bulk phase. This loading volume would fit entirely within **1**'s theoretical helium-accessible pore volume of 0.35 mL/g.

For the simultaneous insertion polymerization experiments, solutions containing the initiator, **S**, and **NVF** were employed. During the initial monomer loading cycle, neat **S** served as the "solvent," which was evaporated at a pressure no lower than 0.7 times its bulk vapor pressure, thus allowing **S** and **NVF** to be loaded into the MOF up to the micropore capacity. In subsequent cycles, acetone was used, ensuring its evaporation took place at pressures above **S**'s vapor pressure but below that of acetone to allow redistribution of monomers without changing their quantity. The loading amount of **NVF**, which is a high-boiling liquid, was directly determined by the amount provided in the feed.

Various initiators, including AIBN, benzoyl peroxide, and ammonium persulfate, were evaluated, and the conditions resulting in the greatest conversion to polymer were adopted (Supplementary Table 1). The heating protocols were adapted based on the decomposition temperature of each initiator, typically involving an initial lower-temperature step to initiate the reaction, followed by a higher-temperature step to improve chain mobility within the pores and enhance conversion (Supplementary Tables 1 and 3). This two-step approach minimized the desorption of monomer by allowing non-volatile oligomers to form inside **1** before ramping to higher temperatures.

Representative examples of the above procedures are provided in Supplementary Method 1. Monomer loading and conversion were monitored in all cases by $^1$H NMR of samples dissolved pre- and post-polymerization. Further details are available in Supplementary Tables 1–3.

## Stepwise polymerization in 1

Preparation of **1⊃PS** was carried out by insertion and polymerization of **S** in **1** in the same manner as described above. Following polymerization, any volatile (non-macromolecular) residues were removed by evaporation at 150 °C for 16 h, and the presence of available micropores was confirmed by sorption measurements. The process was then repeated for insertion and polymerization of **NVF**, with **1⊃PS** used in place of **1** as adsorbent to give **1⊃PS/PNVF**. Monomer loading and conversion were monitored by $^1$H NMR of samples dissolved pre- and post-polymerization in both the **S** and **NVF** steps, as detailed in Supplementary Table 4.

## Synthesis of alternating polymer networks

Alternating polymer networks were synthesized by the same procedure as stepwise polymerization above, with **1** substituted for $1_{x0.10}$, giving **1⊃PS×PNVF** as the final product. In this case, both monomer and dvip loading and conversion were monitored by $^1$H NMR pre- and post-polymerization in each step, as detailed in Supplementary Table 6.

## Data availability

All data supporting the findings of this study are available within the paper, its supplementary information, and source data files. All data are available upon request. The. Source data are provided with this paper.

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

## Acknowledgements

This work was supported by JSPS KAKENHI Programs, Grant Numbers: JP24KJ0710 (K.B.), JP24K01535 (N.H.), JP24K21817 (N.H.), and JP21H04687 (T.U.). This work was also supported by a JST FOREST Program, Grant Number JPMJFR232H (N.H.), and a JST Adopting Sustainable Partnerships for Innovative Research Ecosystem Program (ASPIRE), Grant Number JPMJAP2315 (T.U.).

## Author contributions

Conceptualization: K.B., N.H., and T.U.; supervision: N.H. and T.U.; Experiment and analysis: K.B. and N.H.; Writing: K.B., N.H., and T.U.; Review and editing: N.H. and T.U.

## Competing interests

The authors declare no competing interests.
