## [Transparent Peer Review file · Nature Communications]

Sorting polymerization in a bichannel metal-organic framework

Corresponding Author: Professor Nobuhiko Hosono

Version 0:

Reviewer comments:

Reviewer #1

(Remarks to the Author)

This article reported a monomer sorting polymerization strategy using a famous bi-channel MOF template STAM-1, which has two distinct hydrophilic and hydrophobic pore surfaces. By tailoring the pore dimensions and surface functionalities, they achieved selective uptake of two different monomers into separate channels. This enables the sorting of different vinyl monomers and their in-situ parallel homo-polymerization within the respective channels. Additionally, the introduction of inter-chain cross-linking allowed for the isolation of the binary polymer array by removing the MOF template. This manuscript may be suitable for Nature Communications after addressing the following issues.

- 1) The BET surface of the titled materials should be estimated with N₂ at 77K and CO₂ at 195 K, and made a comparison with the template material STAM-1.
- 2) Temperature-dependent water adsorption/desorption isotherms must be measured to quantify the enthalpy of water adsorption.
- 3) Another experimental evidence such as in-situ FTIR need to be performed to identify the preferential adsorption sites on the two distinct hydrophilic and hydrophobic pore surfaces.
- 4) The authors claimed that incorporating a cross-linking ligand into the MOF structure then enabled the formation of a crosslinked alternating polymer network that remained intact even after the MOF template was removed, resulting in an insoluble alternating array of distinct polymer chains. It is necessary to assess the adsorption/separation performance of this material.
- 5) The authors need to discuss and explain the different H₂O gate-opening properties between titled materials 1, 1 ⊃ PS, 1 ⊃ PS/PNVFstep and 1 ⊃ PS/PNVFsimul.

Reviewer #2

(Remarks to the Author)

Beamsley, Hosono, and Uemura report their investigations into using metal-organic frameworks (MOF) to template polymerizations. The authors cleverly select a MOF that contains pores of different sizes and chemical interactions (one hydrophobic channel, one hydrophilic channel). The authors show that they can selectively load specific monomers to the respective pores (e.g., styrene into the hydrophobic pores) and subsequently polymerize within the MOF. They then show that two distinct polymerizations can occur within the MOF, synthesizing two homopolymers of the chosen monomers (which is impressive). Finally, they use a MOF ligand that can be incorporated into the polymers. Upon removal of the MOF, a copolymer composite is that is quite novel is achieved. Overall, this is fantastic paper and would be an excellent addition to Nature Communications. I have very few suggestions for the authors, the study is well-conducted, the manuscript reads well, and they address many questions that arise (e.g., evidence that there is minimal copolymerization of the two monomers). It could be helpful to add some discussion about the polymer composites at the end, they are quite unique and it's challenging to envision what they would be used for besides being interesting.

Reviewer #3

(Remarks to the Author)

In this work, the authors propose and achieve the selective adsorption of two different monomers into two separate channels of a MOF (one hydrophobic and the other hydrophilic), and, most importantly, they succeed in the polymerization of these monomers to obtain the corresponding polymers. First of all, I must say that the strategy and the results achieved are

exceptional and undoubtedly deserve to be published in such a prestigious journal as Nature Communications.

The work is rigorous. All compounds have been reliably characterized, and the article is well written and excellently presented. Therefore, I recommend that it be accepted as it is. I only wonder whether the authors considered using SCXRD to determine the structure of any of the host–guest aggregates (although I understand this is highly challenging). Maybe some references of single crystal XRD on MOF with guests embedded would be welcome. On the other hand, although it is widely accepted, I do not believe it is strictly accurate to assume that a reduction of X in the amount of adsorbed N₂ necessarily implies an equal numerical reduction in the accessible pore space. Apart from that, I reiterate the excellence of the work and congratulate the authors on the results achieved—results that many of us working in the same field have attempted without success.

Version 1:

Reviewer comments:

Reviewer #1

(Remarks to the Author)

I'm satisfied with the revision made by the authors and suggest the publication of this work in this famous journal.

Reviewer #3

(Remarks to the Author)

The authors have addressed the points raised by me (and I guess also those of other reviewers). Therefore I consider it can be accepted now.

Point-to-Point Responses to Reviewer's Comments

We thank the Reviewers for their useful and encouraging comments on our manuscript. We have carefully considered all concerns raised by the reviewers and have addressed them in the following. Please find attached the revised manuscript with all changes highlighted in dark red. We believe the changes improved the quality of our work to be qualified for publication in *Nature Communications*.

For Reviewer #1:

This article reported a monomer sorting polymerization strategy using a famous bi-channel MOF template STAM-1, which has two distinct hydrophilic and hydrophobic pore surfaces. By tailoring the pore dimensions and surface functionalities, they achieved selective uptake of two different monomers into separate channels. This enables the sorting of different vinyl monomers and their in-situ parallel homo-polymerization within the respective channels. Additionally, the introduction of inter-chain cross-linking allowed for the isolation of the binary polymer array by removing the MOF template. This manuscript may be suitable for *Nature Communications* after addressing the following issues.

- 1) The BET surface of the titled materials should be estimated with N₂ at 77K and CO₂ at 195 K, and made a comparison with the template material STAM-1.

=>We have determined the BET surface areas of **1**, **1**▷PS, **1**▷PS/PNVF_{step}, and **1**▷PS/PNVF_{simul} based on N₂ and CO₂ adsorption isotherms measured at 77 K and 195 K, respectively. The results are summarized in Review-Only Table 1. The N₂-based BET surface area (S_{BET}) of empty **1** was measured to be 288.5 m²/g, which is somewhat higher than other values reported for the STAM series (e.g. 207 m²/g for STAM-1) (Niščáková *et al.*, *Sci. Rep.* **14**, 9232 (2024)). This reflects our choice of the less bulky 5-methylisophthalic acid as ligand to ensure sufficiently large hexagonal channels to incorporate a PS chain, which might not fit in the smaller channels of other STAM-series MOFs. The CO₂-based S_{BET} for empty **1** was found to be 271.0 m²/g, which is consistent with the N₂-based value within a margin of less than 10%.

Sławek *et al.* and Niščáková *et al.* have respectively reported that CO₂ is exclusively adsorbed in the hexagonal pores of STAM-1 (Sławek *et al.*, *Chem. Mater.* **30**, 5116–5127 (2018); Niščáková *et al.*, *Sci. Rep.* **14**, 9232 (2024)), suggesting a similar adsorption behavior to that of N₂. Consistent with this, the PS-loaded samples exhibited a noticeable decrease in S_{BET} for both N₂ and CO₂, reflecting the occlusion of the hexagonal pores by the polymer chains. After PNVF polymerization within the triangular pores, the composites **1**▷PS/PNVF_{step} and **1**▷PS/PNVF_{simul} showed no significant further decrease in S_{BET} for either N₂ or CO₂, as both gases primarily probe the hexagonal pores and do not access the triangular pores.

Thus, N₂ and CO₂ yield comparable results when used as probe molecules for the empty **1**

system. However, differences arise upon filling **1** with polymer - while N₂-derived values draw close to 0 m² g⁻¹ when PS is loaded, CO₂-derived values tend to approach a lower limit of around 145 m² g⁻¹ – a difference we believe to be due to the difference in permeability of the incorporated polymer with respect to each adsorbate. As a single elongated chain of PS is not perfectly dense (Uemura *et al.*, *J. Am. Chem. Soc.* **130**, 6781–6788 (2008)), nor can its packing in the pore be expected to be perfectly space-filling, we expect there to be residual space within the polymer-occupied pores even at maximum loading, and it may be that CO₂ can access this space whereas N₂ cannot due to differences in measurement temperature and kinetic diameter of the two adsorbates. While this means CO₂ more accurately reflects the *absolute* void volume remaining in the pores, we opt to use N₂-derived values in the main text as their 0-intercept property makes S_{BET} (vs. empty **1**) roughly the inverse of polymer loading (% of maximum), and avoids possible confusion as to whether some pores are left unfilled, or if all are filled but with adsorbate-permeable polymer.

Review-Only Table 1. BET surface area (S_{BET}) of **1** and its polymer composites measured using N₂ and CO₂ as a probe gas.

Sample	N ₂ based S_{BET} (m ² g ⁻¹) ^a	CO ₂ based S_{BET} (m ² g ⁻¹) ^b
1	288.5	271.0
1 ⊃PS	67.7	171.1
1 ⊃PS/PNVF _{step.}	27.3	168.3
1 ⊃PS/PNVF _{simul.}	19.9	146.2

^aCalculated based on N₂ adsorption isotherm obtained at 77 K.

^bCalculated based on CO₂ isotherm obtained at 195 K.

=>In response to this comment, we have added the following descriptions about BET surface area of **1** in the Methods section with citation to relevant literature for STAM-1.

(Methods, page 22)

“The Brunauer–Emmett–Teller (BET) surface area of **1**, calculated from the N₂ adsorption isotherm at 77 K, was determined to be 288.5 m²/g. This value is somewhat larger than the reported BET surface area for the related framework STAM-1, supporting the larger size of hexagonal channels in **1** which allow for incorporation of large guests such as **S**.⁵¹”

(Reference)

“51. Niščáková, V. *et al.* Novel Cu(II)-based metal–organic framework STAM-1 as a sulfur host for Li–S batteries. *Sci. Rep.* **14**, 9232 (2024).”

2) Temperature-dependent water adsorption/desorption isotherms must be measured to quantify the enthalpy of water adsorption.

=>We conducted water vapor adsorption measurements for **1**, **1**▷**PS**, **1**▷**PS/PNVF**_{step}, and **1**▷**PS/PNVF**_{simul} at three different temperatures (10 °C, 25 °C, and 40 °C) and calculated the isosteric heat of adsorption (Q_{st}). The Q_{st} value for water in **1** was determined to be approximately 50 kJ/mol, which is consistent with the value reported for STAM-1 (Mohideen, *et al.*, *Nat. Chem.* **3**, 304–310 (2011)), further supporting the similarity between **1** and STAM-1 in their chemisorptive behavior towards water. Moreover, a ‘peak’ in the Q_{st} value was generally observed corresponding to the increasing amounts of hydrogen bonding as uptake reaches saturation of the hydrophilic channel – an effect also reported for STAM-1 (Mohideen, *et al.*, *Nat. Chem.* **3**, 304–310 (2011)). The position of this peak was barely changed upon incorporation of **PS**, but shifted to considerably lower loadings upon incorporation of **PNVF**, backing up our finding that **PNVF** fills this channel whereas **PS** does not.

While Q_{st} values for **1** and **1**▷**PS** were observed in the range of 40–50 kJ/mol, those for **1**▷**PS/PNVF**_{step} and **1**▷**PS/PNVF**_{simul} were slightly lower overall, ranging from 35–45 kJ/mol. Combined with our discussion of H₂O coordination and gate-opening/closing behavior in response to points (3) and (5) below, we believe this change to be due to preoccupation of Cu(II) coordination sites by **PNVF**, preventing the higher enthalpy gain normally experienced by H₂O due to coordinate bond formation upon adsorption to empty **1**.

=>In response to this comment, we have included the following description in the Results section and provided a Q_{st} plot as a function of vapor uptake in the Supplementary Information (Supplementary Fig. 18) alongside the sorption isotherms used to fit Q_{st} (Supplementary Figs. 17) and the FT-IR data discussed below under point (3).

(Results, page 13)

“Isosteric heats of H₂O adsorption further supported site-selectivity of polymers for their respective pores (Supplementary Figs. 17 and 18), and variable-temperature FT-IR spectra point to coordination of **PNVF**’s amide groups to Cu in the triangular pores as a driving force for this selectivity (Supplementary Fig. 19).”

(SI, pages 18 and 19)

Supplementary Fig. 17. H₂O adsorption isotherms measured at 10 °C (circles), 25 °C (squares), 40 °C (diamonds) and their linear interpolants (lines) for (a) **1**, (b) **1▷PS**, (c) **1▷PS/PNVF_{step.}**, (d) **1▷PS/PNVF_{simul.}**. These isotherms were used for the determination of the isosteric enthalpy of adsorption of water.

Supplementary Fig. 18. Isosteric heat of adsorption (Q_{st}) of H_2O derived by fitting of the Clausius-Clapeyron equation for sorption isotherms of (a) **1**, (b) **1⊃PS**, (c) **1⊃PS/PNVF_{step}**, (d) **1⊃PS/PNVF_{simul.}**. Fits were carried out only for uptake values where data could be interpolated for all three temperatures (10, 25, and 40 °C). A sharp peak originating from increased hydrogen bond formation is observed at the uptake where triangular pores become saturated, corroborating that the change in available volume in these pores upon introduction of PS is negligible ($164 \rightarrow 159 \text{ cm}^3 \text{ (STP)/g}$), but is large for PNVF ($164 \rightarrow 60 \text{ cm}^3 \text{ (STP)/g}$). This supports the site selectivity of the two species.

3) Another experimental evidence such as in-situ FTIR need to be performed to identify the preferential adsorption sites on the two distinct hydrophilic and hydrophobic pore surfaces.

=>We thank the reviewer for this valuable suggestion. In response, we conducted diffuse reflectance FT-IR (DRIFTS) measurements from 50 to 150 °C on **1**, the composites (**1**⊃**PS**, **1**⊃**PS/PNVF**_{step}, and **1**⊃**PS/PNVF**_{simul}), and the corresponding homopolymers (PS and PNVF). The diffuse reflectance FT-IR spectra are presented in Supplementary Fig. 19. Unfortunately, due to significant overlap of characteristic amide (–N–H bend) and carbonyl (C=O stretch) peaks from PNVF, as well as C=C stretches from PS, with the carboxylate (COO⁻) bands of **1**, meaningful spectral analysis of the polymers themselves proved to be difficult as the MOF peaks were overwhelmingly dominant. Nevertheless, IR analysis provided some valuable insights through shifts in the peaks of **1** itself, in conjunction with observations of the non-overlapping bands from PNVF (N-H stretch) and H₂O (O-H stretch).

We believe that the inclusion of styrene in the hexagonal pores is primarily driven by physisorption via van der Waals interactions. In contrast, the inclusion of polar monomers likely involves coordination-driven interactions. Specifically, the Lewis basic amide group of NVF may coordinate to the Lewis acidic Cu(II) open metal sites (OMS) present in the paddle-wheel clusters of **1**. Such coordination behavior is well-documented in Cu-based MOFs, e.g. STAM-1 (McKellar, *et al.*, *Nanoscale* **6**, 4163–4173 (2014)) and HKUST-1 (e.g. Kim, *et al.*, *J. Am. Chem. Soc.* **137**, 10009–10015 (2015)). While the peak overlaps described above limited our observations of polymer-side bands to a slight blue-shift in PNVF's N-H stretching band, we were able to see more profound changes on the MOF side.

Without polymer, **1** begins in gate-open (**1**⊃H₂O) phase, with H₂O's O-H stretching bands clearly visible in the 3300-3700 cm⁻¹ region. Upon heating to 150 °C under N₂, H₂O completely desorbs as evidenced by the complete disappearance of O-H bands. The loss of axially coordinated water also causes electronic changes to the coordination environment around Cu(II), causing shifts in the bands of equatorial COO⁻ groups, along with emergence of a new peak at 1513 cm⁻¹ which is seemingly unique to the gate-closed phase. Intriguingly, while **1**⊃**PS** exhibits near-identical changes to **1** when heated, **1**⊃**PS/PNVF** samples do not. These largely exhibit the same COO⁻ bands as **1**⊃H₂O, but do so all the way up to 150 °C, despite the complete disappearance of H₂O as evidenced by the loss of O-H bands. This suggests that the same effects on the Cu(II) coordination environment as seen in the H₂O-coordinated phase are present even in absence of H₂O when PNVF is incorporated, strongly indicating coordination of PNVF to the complex.

The stabilization of the open-phase structure of **1** by PNVF inclusion, as seen in the disappearance of gate-closing transitions from VT-XRD when PNVF is incorporated (Supplementary Figs. 8-11), also supports NVF coordination to the Cu OMS. It is well-established that guest coordination to the Cu sites is essential for inducing the closed-to-open phase transition of the STAM series MOFs as discussed in our response to point (5) below. This observation, combined with known coordination chemistry, underpins our interpretation of monomer-specific interactions within the MOF.

(SI, page 20)

Supplementary Fig. 19. Variable-temperature diffuse reflectance FT-IR (DRIFTS) spectra of composites 1DPS , $1\text{DPS/PNVF}_{\text{step}}$, $1\text{DPS/PNVF}_{\text{simul}}$, PS, and PNVF. (a) Initial spectra at 50 °C. (b) Final spectra at 150 °C. Both the gate-open ($1\text{D}\text{H}_2\text{O}$ at 50 °C) and gate-closed (1 at 150 °C) forms of the MOF are provided for comparison. Magnified views are provided for comparison of (c) the N-H stretching band in bulk PNVF vs. 1DPS/PNVF composites, and (d) COO^- stretches of various composites of 1 .

Note that while H_2O is present in all composites at 50 °C, these are fully dehydrated by 150 °C as evidenced by the loss of O-H stretching bands. Compounds which transition to the gate-closed phase additionally experience a red shift of $\nu_{\text{as}}\text{COO}^-$ and display an additional

characteristic band at 1513 cm^{-1} upon dehydration. Composites containing PNVF exhibit similar spectra to the H_2O -coordinated form of **1**, even when dehydrated, indicating a similar axially-coordinated environment around Cu and, coupled with a shift in the N-H band, show that PNVF's amide moiety coordinates to Cu in place of the typical H_2O . This explains both the self-sorting behavior of NVF and the "gate-locking" behavior in composites of **1** containing PNVF (Supplementary Figs. 8-11).

4) The authors declaimed that incorporating a cross-linking ligand into the MOF structure then enabled the formation of a crosslinked alternating polymer network that remained intact even after the MOF template was removed, resulting in an insoluble alternating array of distinct polymer chains. It is necessary to assess the adsorption/separation performance of this material.

=>To assess the intrinsic porosity of the crosslinked alternating polymer network (**PS×PVAm**), we conducted N_2 and CO_2 adsorption measurements at 77 K and 195 K, respectively. We also conducted vapor sorption measurements of H_2O at 298 K. Please refer to the following Review-Only Figure 1. The material exhibited negligible N_2 and CO_2 gas uptake across the entire pressure range, indicating that it is essentially non-porous in the dry state. We believe this observation is reasonable, as removal of the host MOF likely allows the polymer network to shrink, collapsing into the void spaces previously occupied by the framework. The network appears sufficiently flexible to adopt a more densely packed conformation upon template removal.

However, in solution, the polymer network can swell with solvent molecules, resulting in a particle size comparable to that of the original composite ($\mathbf{1}_{x0.10} \rightarrow \mathbf{PS}\times\mathbf{PVAm}$), as seen in Fig. 5d of the main manuscript. This is corroborated by our H_2O vapor sorption results which exhibit uptake with a high level of hysteresis characteristic of swelling in hydrogen-bonding polymers (Chen, *et al.*, *Nat. Commun.* **9**, 3507 (2018)), which is somewhat expected behavior considering previous studies also show water uptake in PVAm homopolymers (Casadei, *et al.*, *Membranes* **9**, 119 (2019)). This suggests that **PS×PVAm** forms a flexible network with transient, solvent-accessible pores in the swollen state, but does not exhibit permanent microporosity or gas separation properties characteristic of conventional rigid porous materials.

Importantly, we emphasize that the primary objective of this work is not the synthesis of porous polymers, but rather the development of a novel nanomaterial fabrication strategy. Through spontaneous monomer-sorting and in situ polymerization within a bichannel MOF, we have achieved the formation of alternating polymer networks directly from monomer mixtures in a single step. We believe this represents a significant advancement in synthetic methodology, independent of the final material's porosity. Moreover, we acknowledge that MOF-templated methods for synthesizing porous crosslinked polymers have been previously reported (e.g. Kobayashi, *et al.*, *ACS Appl. Mater. Interfaces* **9**, 11373–11379

(2017)). Therefore, the porosity of the product is beyond the scope of the present study, and we do not consider specific adsorption or separation performance to be a relevant characteristic of the alternating polymer network synthesized herein.

Review-Only Figure 1. Adsorption (circles) and desorption (crosses) isotherms of PS×PVAm. (a) N₂ at 77 K, (b) CO₂ at 195 K, (c) H₂O at 298 K.

5) The authors need to discuss and explain the different H₂O gate-opening properties between titled materials **1**, **1**⊃PS, **1**⊃PS/PNVF_{step} and **1**⊃PS/PNVF_{simul}.

⇒ We understand that the reviewer is referring to the differences in the H₂O adsorption isotherms observed for **1**, **1**⊃PS, **1**⊃PS/PNVF_{step}, and **1**⊃PS/PNVF_{simul}, as shown in Supplementary Fig. 16, and how they relate to phase transitions between gate-open and gate-closed forms of **1** as investigated using VT-XRD in Supplementary Figs. 8-11. As discussed in the main text, **1** exhibits gate-opening behavior, which is associated with guest coordination to the axial open metal sites (OMSs) on the Cu paddle-wheel clusters. These sites are oriented toward the interior of the triangular pores, and thus, the gate-opening behavior is driven by coordinative guest occupation within these pores – a role typically

filled by H₂O as discussed in length by McHugh and co-workers for STAM-17-OEt (McHugh, *et al.*, *Nat. Chem.* **10**, 1096–1102 (2018)).

Both empty **1** and **1**⊃**PS** display typical gate-type (sigmoidal) H₂O adsorption isotherms (Supplementary Fig. 16), as the Cu OMSs in the triangular pores remain accessible. This observation supports the proposed structure of **1**⊃**PS**, where the triangular pores are unoccupied by PS chains and remain available for H₂O adsorption, which drives the gate-opening phase transition. Additionally, both **1** and **1**⊃**PS** exhibit temperature-dependent gate-closing behavior (Supplementary Figs. 8 and 9), with pre-coordinated H₂O molecules being released at elevated temperatures (above 100 °C), inducing a transition to the closed phase. This change was also observed in the VT-IR described under point (3), confirming loss of H₂O by the disappearance of O-H stretching bands.

In contrast, **1**⊃**PS**/**PNVF**_{step.} and **1**⊃**PS**/**PNVF**_{simul.} exhibit significantly suppressed H₂O adsorption and do not display the characteristic gate-type isotherms (Supplementary Fig. 16), instead displaying Type I curves typical of adsorption on rigid frameworks and bulk surfaces. This is attributed to the occupation of the triangular pores by PNVF chains, which reduces H₂O vapor uptake and, moreover, causes both composites adopt a permanently open-phase structure due to the coordination of PNVF chains to Cu sites within the triangular pores as was supported by VT-IR. This coordination stabilizes the open phase and prevents structural transitions, even under elevated temperatures where the H₂O vapor typically required to stabilize this form is lost (Supplementary Fig. 11). These observations, together with supporting data, confirm the respective locations of PS and PNVF within the MOF.

=>In response to this comment, we added the following discussions in the caption of Supplementary Fig. 16.

(SI, page 17)

“Regarding the H₂O adsorption profiles, the gate-opening behavior, signified by a sigmoidal isotherm profile, is observed in (a) **1** and (b) **1**⊃**PS**, but not for (c) **1**⊃**PS**/**PNVF**_{step.} and (d) **1**⊃**PS**/**PNVF**_{simul.} The closed-to-open transformation of **1** is induced by the coordination of H₂O molecules to the Cu open metal sites located within the triangular pores of the MOF.¹ In **1**⊃**PS**, the triangular pores remain accessible, as PS chains occupy only the hexagonal pores. Thus, this composite shows gate-opening behavior as seen in empty **1**. In contrast, both **1**⊃**PS**/**PNVF**_{step.} and **1**⊃**PS**/**PNVF**_{simul.} exhibit significantly reduced H₂O uptake and do not display typical gate-type isotherms, due to the occupation of the triangular pores by PNVF chains. The coordination of PNVF to the Cu open-metal sites stabilizes the open-phase structure, preventing coordination of H₂O and disabling dynamic gate-opening behavior by keeping **1** open permanently.”

For Reviewer #2:

Beamsley, Hosono, and Uemura report their investigations into using metal-organic frameworks (MOF) to template polymerizations. The authors cleverly select a MOF that contains pores of different sizes and chemical interactions (one hydrophobic channel, one hydrophilic channel). The authors show that they can selectively load specific monomers to the respective pores (e.g., styrene into the hydrophobic pores) and subsequently polymerize within the MOF. They then show that two distinct polymerizations can occur within the MOF, synthesizing two homopolymers of the chosen monomers (which is impressive). Finally, they use a MOF ligand that can be incorporated into the polymers. Upon removal of the MOF, a copolymer composite that is quite novel is achieved. Overall, this is fantastic paper and would be an excellent addition to Nature Communications. I have very few suggestions for the authors, the study is well-conducted, the manuscript reads well, and they address many questions that arise (e.g., evidence that there is minimal copolymerization of the two monomers). It could be helpful to add some discussion about the polymer composites at the end, they are quite unique and it's challenging to envision what they would be used for besides being interesting.

=>We are grateful for the encouraging comments and constructive feedback on our work. We believe that the ability to control polymer arrangement at the single-chain level represents a significant breakthrough in the design of nanostructured devices, combining polymers' flexibility with crystals' structural precision and reliable performance, particularly for applications in molecular electronics and energy conversion technologies. For instance, an interdigitated, alternating arrangement of donor and acceptor polymers is considered an ideal architecture (so-called ordered heterojunction structure) for thin-film organic solar cells; however, such molecular-level control has remained unattainable using conventional fabrication techniques. By leveraging the bichannel MOF-based sorting polymerization developed in this study, it becomes possible to construct such optimized architectures. While our work demonstrates this approach using common vinyl polymers, the same strategy can be extended to conductive polymers, thereby enabling the formation of molecular-level alternating arrays of electronically distinct polymer species.

=>In response to this comment, we have added the following descriptions in the main text.

(Discussion, page 18)

“Further, the ability to control polymer arrangement at the single-chain level represents a breakthrough in the construction of nanostructured devices, combining the softness of polymers with the sub-nanoscale regularity typically reserved for crystalline systems, which is especially relevant for applications in molecular electronics and energy conversion technologies.”

For Reviewer #3:

In this work, the authors propose and achieve the selective adsorption of two different monomers into two separate channels of a MOF (one hydrophobic and the other hydrophilic), and, most importantly, they succeed in the polymerization of these monomers to obtain the corresponding polymers. First of all, I must say that the strategy and the results achieved are exceptional and undoubtedly deserve to be published in such a prestigious journal as Nature Communications.

The work is rigorous. All compounds have been reliably characterized, and the article is well written and excellently presented. Therefore, I recommend that it be accepted as it is. I only wonder whether the authors considered using SCXRD to determine the structure of any of the host–guest aggregates (although I understand this is highly challenging). Maybe some references of single crystal XRD on MOF with guests embedded would be welcome. On the other hand, although it is widely accepted, I do not believe it is strictly accurate to assume that a reduction of X in the amount of adsorbed N_2 necessarily implies an equal numerical reduction in the accessible pore space. Apart from that, I reiterate the excellence of the work and congratulate the authors on the results achieved—results that many of us working in the same field have attempted without success.

1) Discussion about the single-crystal XRD of the guest-loaded MOF.

=>We thank the reviewer for the encouraging comments. In this work, the choice to work with powder techniques was a strategic decision taken early in the project. While single-crystal X-ray diffraction (SXRD) analysis has proven to be a powerful technique for the structural analysis of small molecular guests in MOFs with atomic precision, the nature of polymers negates much of the advantage SXRD holds over powder analysis. In addition to inherent structural irregularities such as the presence of terminal groups, sequence isomerism, and mixed stereoisomerism (tacticity), any polymer of appreciable molecular weight inevitably would have to extend through multiple unit cells of the parent MOF's crystal structure, unlike smaller guest molecules which can often be localized to specific sites in the cell. These factors make atomic-precision SXRD of polymer guests impossible in all but the most exceptional circumstances (i.e. those in which the isomerism, conformation, and orientation of polymer chains are consistent throughout all unit cells, and the unit cell length in the channel direction is an exact integer multiple of the polymer's repeat unit). In light of this, we concluded that the best data one could hope for in our system would be a disordered cloud of electron density in each polymer-occupied pore, and that this level of resolution could be sufficiently obtained using powder techniques.

Thus, we relied on PXRD pattern refinement to localize electron density associated with the polymeric guests, which provided sufficient evidence to support the conclusions of our study. While SXRD is indeed a powerful tool for studying small molecular guests, as demonstrated in several reports on STAM-1 (e.g., McKellar, *et al.*, *Nanoscale* **6**, 4163–4173 (2014)), its application to larger, disordered polymer guests remains highly challenging.

=>In response to this comment, we have added the following description to the main text, with citation to relevant SXRd-based studies on small guest molecules in STAM-1.

(Results, page 6)

“Furthermore, single-crystal X-ray diffraction analysis of solvent-loaded STAM-1 has demonstrated that solvent molecules bearing coordinative moieties preferentially interact with the axial coordination sites of the paddle-wheel clusters, highlighting the potential for pore-selective inclusion of specific guests through coordination-driven interactions.³⁴”

(Reference)

“34. McKellar, S. C. *et al.* The effect of pressure on the post-synthetic modification of a nanoporous metal–organic framework. *Nanoscale* **6**, 4163–4173 (2014).”

2) Discussion about the relationship between gas adsorption amount and actual pore volume.

=>We thank the reviewer for their valuable input. We acknowledge that while *changes* in gas sorption amount are presumably proportional to available pore volume, the correlation is not strictly 1-to-1 due to the nonzero size of adsorbate molecules. Indeed, different adsorbates with different kinetic diameters may indicate different adsorption volumes (and consequently different S_{BET}) depending on the range of pore sizes their kinetic diameter is capable of probing, as seen in our CO₂ results in the response to Reviewer 1 above.

Moreover, under our target conditions that there only be one polymer chain per pore, the linearly extended conformation of a single polymer chain can be expected to be imperfectly dense and imperfectly packed (there is space between the pendant groups, as well as between the chain and the pore wall). We can thus expect the minimum pore volume (at maximum loading, i.e. 100% of pores have a polymer chain in them) to be nonzero in cases where adsorbate is capable of permeating the ‘free space’ within the polymer chain. This may explain the convergence of our CO₂ results to a nonzero lower bound instead of to zero.

We elected to present N₂-based void volumes as projections of the residual void in hexagonal pores due to the property that this adsorbate’s sorption capacity converges toward zero as polymer loading approaches its maximum, making the residual sorption capacity roughly proportional to the amount of space that may still be filled with polymer – that is, the N₂-derived residual space is more reflective of the space available for monomer/polymer to fill. In terms of absolute pore volume, smaller and more mobile adsorbates can, of course, still fill interstitial spaces in and around the polymer chains - we find that CO₂ is a more reliable probe of this full space, retaining around half of its sorption capacity even at maximum polymer loading, consistent with the low density of single PS chains in MOF channels as reported previously (Uemura *et al.*, *J. Am. Chem. Soc.* **130**, 6781–6788 (2008)). However, we stress that this residual void does not imply the presence of less than one chain per channel (i.e. sub-maximum loading), but rather that all pores have a chain which is not

perfectly space-filling in nature, and elect to use N_2 -accessible voids in the main text to avoid this confusion.

We realize that the wording “Residual void %” in Fig. 3 may cause misunderstandings as the adsorbate-accessible void volumes are not necessarily equal to the absolute void volume. To rectify this we have changed the wording of this figure to reflect that the values in question represent N_2 and H_2O -*accessible* voids, respectively.

(Fig. 3, revised)

Response to Editorial Requests:

Characterization data.

=>We have added the characterization data of dvpH₂ in the Supplementary Information.

(SI, page 35)

¹³C NMR (125 MHz): δ (ppm) 169.1, 136.5, 135.2, 135.0, 133.9, 128.5, 118.9, 117.1; HRMS (ESI-negative): calcd. for C₁₂H₉O₄ [M – H][–]: m/z = 217.05063; found 217.05240.

Solution-state ¹H and ¹³C nuclear magnetic resonance (NMR) spectra were recorded using a Bruker Avance III HD spectrometer equipped with a PABBO probe operating at 500 MHz. Electrospray ionization time-of-flight mass spectrometry (ESI-MS) was performed on a Bruker Q-TOF mass spectrometer with MeOH as eluent.”

Editorial Policy Checklist

=>We have enclosed the Editorial Policy Checklist.

Data Availability Statement

=>We have updated the Data Availability Statement as follows.

“All data generated in this study are provided in the Supplementary Information/Source Data file. All data are available upon request. Source data are provided in this paper.”

Source Data

=>We have uploaded the Source Data zip file, which includes Excel files for each figure.